# The Fate of Secondary Electrons in Water upon High-Energy Electron Impact: Changes in the Presence of Nanobubbles

**DOI:** 10.3390/ijms26188909

**Published:** 2025-09-12

**Authors:** Yulia V. Novakovskaya, Nikolai F. Bunkin, Sergey A. Tarasov, Natalia N. Rodionova, Anastasia O. Petrova, German O. Stepanov

**Affiliations:** 1Chemistry Department, Lomonosov Moscow State University, Leninskie Gory, 1/3, 119991 Moscow, Russia; 2Department of Fundamental Sciences, Bauman Moscow State Technical University, 2nd Baumanskaya Str. 5, 105005 Moscow, Russia; 3Research and Development Department, OOO “NPF “Materia Medica Holding”, 129272 Moscow, Russia or satarasovmail@yandex.ru (S.A.T.); rodionovann@materiamedica.ru (N.N.R.); petrovaao@materiamedica.ru (A.O.P.); or stepanov_go@rsmu.ru (G.O.S.); 4Department of Molecular and Cellular Pathophysiology, Institute of General Pathology and Pathophysiology, 125315 Moscow, Russia; 5Institute of Biomedicine, Pirogov Russian National Research Medical University, Ministry of Health of the Russian Federation, 117513 Moscow, Russia

**Keywords:** gas nanobubbles, water, aqueous solutions, hydrated electron, stabilization energy, electron detachment energy, electron affinity, UV radiation absorption, kinetic schemes

## Abstract

Electrons localized by water molecules are known as hydrated electrons. The composition of the aqueous environment determines their state and behavior. In this experimental and theoretical work, hydrated electrons were formed in aqueous solutions upon high-energy electron impact, and the dependence of their characteristics on the presence of nanobubbles generated during vibrational treatment was investigated. To explain the results, quantum chemical simulations were carried out, and diverse possible kinetic schemes were considered. Absorbance of deionized water and NaCl aqueous solution was measured at a wavelength of 600 nm, which falls in the range typical of hydrated electrons. The principal differences in the spectral responses of the samples were discovered depending on whether they were preliminarily subjected to repeated vigorous shaking or not. Vigorous shaking caused a noticeable increase in both the integral and maximum absorbance, and the absorbance decay was significantly slower. The effects observed in the vibrationally treated aqueous samples were found to be explained only in the framework of a kinetic scheme that assumes the repeated solvation of electrons, which are transferred from a localized to a delocalized (free) state upon the energy absorption. This repeated solvation is possible only when the secondary electrons are localized on the inner surfaces of the boundary hydration shells of nanobubbles, which are formed in the process of shaking. Thus, nanobubbles substantially change the apparent gross lifetime and properties of hydrated electrons, and these changes, in turn, can indicate the presence of nanobubbles in water and aqueous solutions.

## 1. Introduction

As noted in our previous work [1], when water or an aqueous solution is subject to shaking, the density distribution inside the sample becomes non-monotonic. This is due to local breaks in the continuity of the liquid during shaking; the cavities formed in the discontinuity domains are filled with gas dissolved in the liquid, which leads to the appearance of the bubble phase. The generation of gas bubbles as a result of intense shaking can be considered as hydrodynamic cavitation, see, for example, the recent review [2]. At the same time, bubbles resulting from hydrodynamic cavitation disappear when the external effect, which is the source of cavitation, ceases. As shown in [1] (see Figure 1 in this work), the nanometer-sized bubbles formed as a result of shaking exist during the entire dynamic light scattering experiment, they must be stabilized in some way (otherwise these bubbles would have to collapse within tens of microseconds).

In general, nanobubbles are not resistant against outward diffusion of gas due to the significantly higher gas pressure inside them compared to the atmospheric one, which eventually leads to their collapse [3]. The lifetime of a nanobubble is proportional to its squared radius, and at the nanobubble radius R_b_ = 100 nm is about 0.1 ms [4]. The mechanical and diffusion equilibrium state of such nanobubbles requires a compensation of the surface tension. Equilibrium becomes possible when structure-breaking (chaotropic) anions are localized in the superficial (boundary) layers of the bubbles. This concept has been confirmed by the experimentally discovered increase in the lifetime of gas microbubbles with an increase in the concentration of electrolyte added to aqueous solutions [5]. Notably, even in the absence of intentionally added foreign ions, ions do appear in water due to the dissolution of carbon dioxide and its hydration according to reaction (1):H_2_O + CO_2_ ⇔ H_3_O^+^ + HCO_3_^−^(1)

As shown in our previous paper [1], the content of dissolved carbon dioxide increases upon shaking. Reaction (1) produces bicarbonate anions, which can be located within the boundary trimolecular (in thickness) layer of water molecules around a gas core of the bubble, in this way ensuring the stabilization of the latter. We named such structures bubstons, which is an abbreviation for “bubbles stabilized by ions” [6,7].

The volume density of nanobubbles generated as a result of hydrodynamic cavitation increases at more intense shaking. As a result, significant distortions of the hydrogen-bond (H-bond) network occur with the appearance of multiple defects. This particularly facilitates the dissolution of atmospheric gases in water and aqueous solutions, resulting in the formation of nanobubbles. Such nanobubbles can affect the refractive index [8] and light scattering [9] in the medium, and, if stabilized, are subject to the flotation effect, adsorbing dissolved substances on their surface [7,10].

The proposed structure of the negatively charged spherical shell of the bubston’s gas core [1] allows for the additional inclusion of negatively charged particles, including electrons, provided that the aqueous phase is supersaturated with such particles. These conditions can be created by irradiating the sample with an electron beam. The lifetime of secondary solvated electrons (e_aq_) is about 10 µs [11]. Such electrons, being initially quasi-free and delocalized in the conduction band, are at first transferred into an intermediate state (the so-called wet electrons) within about 110–180 fs, and, in another 240–540 fs they become completely solvated, i.e., belong to the localized state [12,13,14,15,16,17], and referred to as hydrated electrons. The charge distribution of localized electrons is described by an s-function, and the regions of localization are D-type Bjerrum defects of the H-bond network, where two or more water molecules face each other with their OH groups.

According to electron spin echo experiments, an electron in water is surrounded on the average by six molecules [18]. The intermediate (wet) state of the electron corresponds to its excited p-state, which is non-degenerate, and the energy gap between these states fall in a range of 0.25–0.5 eV [19]. The p-state is less localized, and the s → p electron transition determines the characteristic absorption spectrum of hydrated electrons with a maximum at a wavelength of 720 nm [13,20,21,22,23,24,25]. It should also be noted that electrons obtained as a result of water photoionization are localized chiefly at the water-gas interface [26]. Their transition to the bulk water requires overcoming a potential barrier of 0.2–0.5 eV. This means that in the presence of nanobubbles in water, secondary electrons can be localized within their boundary spherical layers. The existence of an energy barrier should increase the lifetime of secondary electrons in comparison with the bulk water. In addition, it can be expected that within the boundary layers of nanobubbles, the character of the electron localization should differ from that in the bulk water, which, in turn, should change their excitation and detachment energies.

Thus, one may expect that the absorbance of radiation by hydrated electrons in water/aqueous salt solution that contains nanobubbles should change. The study of changes in the absorbance of secondary electrons in aqueous solutions after vibrational treatment, as well as the need for a kinetic and quantum chemical description of this process, predetermined the concept of our work. It is aimed at obtaining independent corroboration evidence in favor of the previously extended model of bubstons [6,7], particularly the expected character of the localization of diverse ions within or near their boundary hydration shells, and clarifying the role of such nanobubbles in the processes that take place in water and aqueous solutions. In the latter case, one may expect to observe changes not only in the steady-state characteristics (such as the absorbance spectrum) but also in their time dependences that are predetermined by the kinetics of processes, which proceed in aqueous solutions and may vary depending on the existence and concentration of nanobubbles.

## 2. Results

### 2.1. Spectral Studies

The absorbance decay of a hydrated electron in an aqueous solution is known to be characterized by a microsecond order. However, it remains unclear whether the lifetime and absorbance of a hydrated electron depend on the presence of nanobubbles generated during vibrational treatment of the studied solution. Such time dependences were obtained for pure water and aqueous solutions both in their intrinsic state and upon the preliminary vigorous shaking. The hydrated electrons were generated in the samples as a result of the high-energy electron impact, which caused the formation of diverse radical particles and low-energy secondary electrons (see Section 4.2). As mentioned above, the secondary electrons on a half-picosecond scale become localized, or hydrated. In our experiments, we assessed the dependence of the absorbance of these electrons on time. The general character of the dependences in different samples was the same, with the variations in the slopes of ascending and descending segments of the absorbance signal, its temporal width, and the asymptotic value. Figure 1 shows a typical time dependence of the absorbance of an aqueous NaCl aqueous solution (10 mg/L) at a wavelength of 600 nm upon the high-energy electron impact. In the case of the NaCl solution, as can be seen (in Figure 1), the absorbance is noticeably non-zero within a time interval (τ) of 10 μs.

The sample absorbance is expressed as log(I_in_/I_out_); where I_in_ is the incident radiation intensity, and I_out_ is the transmitted radiation intensity. Typically, the absorbance increases rapidly at the irradiation of the sample with a high-energy electron beam, reaching a maximum within 2–4 μs. This is followed by an apparently exponential decrease in the absorbance, which lasts around 9–12 μs. Both the peak values of such dependences (the maximum value) and the integral ones are of primary interest to us, as well as the rate of increase and decrease in the absorbance depending on the nature of the test sample.

The peak absorbances (at λ = 600 nm) of different samples are compared in Figure 2. As a reference, the spectral data of individual water samples were taken. These data illustrate the changes that take place (i) upon the intensive vibrational treatment of water during sequential dilutions (HD VT Water), (ii) upon the dissolution of NaCl salt (10 mg/L), and (iii) upon the intensive vibrational treatment of the salt solution during its repeated 100-fold dilutions (HD VT NaCl).

As can be seen from Figure 2, the average peak absorbance of the samples subjected to repeated vibrational treatment is approximately 50% higher than that of the reference sample. Furthermore, in the presence of NaCl, the peak absorbance is significantly reduced (by almost 30%). However, upon the repeated vibrational treatment, each step of which is accompanied by a 100-fold (calculated) decrease in the NaCl concentration, it increases almost twice up to the level typical of the vibrationally treated water.

The kinetics of the spectral response is shown in Figure 3, where the decay rates of the absorbance after reaching the peak value are compared for different samples. The peak was reached 2–4 µs after the onset of the high-energy electron impact (6 MeV, a 1.5-µs pulse). At this step, to retrieve tentative quantitative characteristics of the dependences, the following points were taken into account. Most of the decay processes are characterized by the first-order kinetics. The actual decrease in the absorbance with time (see Figure 1) is well linearized in a semilogarithmic frame of reference. For these reasons, the absorbance decay was approximated by an exponential function *A* = *αexp(−βt)*, where *β* factors can be treated as apparent first-order rate constants. These *β* factors are shown in Figure 3.

Note that in the reference sample, as well as in the original NaCl solution, the absorbance decreases significantly faster than in HD VT samples, and the difference between HD VT water and HD VT NaCl is statistically insignificant. The absorbance of the latter samples is non-zero (although small) even at the times when the approximating exponent reaches its asymptotic zero level. Combining these data with the data of the average peak absorbance shown in Figure 2, we can draw a preliminary conclusion that the electron impact leads to the appearance of a larger number of particles. These particles can absorb radiation at a wavelength of 600 nm in the samples that were preliminarily subjected to intense shaking, and the apparent lifetime of these particles is longer.

We know (see above and the data published in [1] that the stronger the vibrational treatment, the larger the number of nanobubbles appeared in the sample. Then, it is natural to suppose that it is their presence in the bulk liquid and, in general, the increasing number of defects in the H-bond network of water that may result in the growth of both the peak and integral absorbances of electromagnetic radiation at a wavelength of 600 nm. In order to clarify the nature and mechanism of the effect one should know the energetic characteristics of particles, either built in the boundary shells of nanobubbles or present in the bulk water, which can absorb energy in the specified range. This information can be retrieved from model simulations.

### 2.2. Nonempirical Simulations: Structural and Energetic Characteristics of Hydrated Electrons and Hydration Complexes of Sodium and Chloride Ions

As follows from our previous paper [1], nanobubbles formed in bulk water upon shaking can be considered as a gas core surrounded with a boundary transient hydration shell. Effective sizes of nanobubbles formed under physical conditions of interest fall in a range of 75 to 175 nm [1]. An accurate estimate of the shell thickness is a challenge, but the spherical layer directly affected by the neighboring gas core can be thought of as involving no less than a dozen water monolayers, which accounts to about 5 nm. The whole nanobubble structure can be stabilized by the ions included in the boundary shell. In the absence of any foreign particles intentionally dissolved in water, these are bicarbonate ions formed upon the hydration of carbon dioxide and separated from the gas core with no less than a monomolecular-thick water layer [1]. To judge the probable domains of the localization of sodium and chloride ions in NaCl solutions and excess secondary electrons formed upon the high-energy electron impact, we consider (see Section 4) diverse clusters, which mimic instantaneous fragments of either the boundary shells of nanobubbles or bulk water and compare the structural and energetic characteristics of the resulting cluster structures. A schematic illustration of a nanobubble immersed in bulk water is shown in Figure 4, where small octagons indicate the regions, to which model systems correspond.

#### 2.2.1. [Na(H_2_O)_n_]^+^ Aqua Complexes

Based largely on the previously obtained results [27] we can state that the typical coordination numbers of the sodium cation are five and six. The ion inside its solvation shell can be located near the center (the so-called bulk structures, Figure 5a) either more or less significantly shifted relative to it (surface structures, Figure 5b). In the surface structure of the largest considered cluster [Na(H_2_O)_51_]^+^ (Figure 5b), the difference in the effective thickness of the hydration shell in its different hemispheres reaches 1.45 Å, which corresponds to almost a monolayer of water molecules. Intermediate cluster configurations are also possible and (as follows from our dynamic calculations [27] play the role of transient structures in the mutual transformations of aqua complexes of the bulk and surface kinds.

The first solvation shells of a sodium ion in the surface structures are more compact (the average Na–O distances are smaller in them). The structures themselves can be characterized as vertically or horizontally elongated bowls (prolate bowls), in the middle region of which the ion is localized, being only partly immersed in the H-bond network of water molecules. Such structures are energetically preferable when the number of water molecules does not exceed 40. For example, for [Na(H_2_O)_28_]^+^ and [Na(H_2_O)_38_]^+^ clusters, the relative energies of the bulk structures are higher than those of the surface ones by 7.4 and 3.1 kcal/mol, respectively. If we turn to the relative enthalpies of the cluster structures normalized to the numbers of water molecules (*H_rel_*/*n*), a difference of 0.3 and 0.1 kcal/mol with respect to the value of 17.7 kcal/mol, which characterizes the surface structures, can be found. Note that at such a number of water molecules, sodium ion turns out to be pentacoordinated in the clusters of both bulk and surface kinds.

At a larger number *n* of water molecules around a sodium ion, the bulk configurations of its aqua complexes are preferable. The ion in these complexes is always hexa-coordinated, whereas in the surface and transient configurations, the number of molecules in its first hydration shell can be either 5 or 6 with a clear trend of increasing as the total number of molecules in the cluster increases. It is worth noting that in smaller clusters (with *n* < 40) the number of H-bonds between water molecules in the bulk structures is typically smaller by one, whereas at *n* > 45 it is usually larger by two in such structures. This reflects changes in the degree of general connectivity of the structure. It is also noteworthy that the relative enthalpy of bulk clusters normalized to the number of water molecules at such their sizes decreases to 17.7 kcal/mol, which is characteristic of the surface structures at a smaller number of molecules, while the *H_rel_*/*n* values of the latter structures increase to an average of 18.1 kcal/mol. Thus, always the most stable cluster structures are characterized by approximately the same energy per water molecule (17.7 kcal/mol).

Obviously, bulk structures are more typical of the bulk water, where ions are relatively uniformly surrounded with water molecules. At the same time, surface structures can serve as prototypes of instantaneous fragments of liquid in the boundary shells of nanobubbles. Here, the numbers of free (not involved in H-bonds) OH groups of the peripheral molecules in the bulk and surface clusters are close, being less by one or two in the surface structures (Table 1). These are the groups that can provide the binding of this cluster to the neighboring water molecules in more extended systems. In the bulk structures, these groups are distributed fairly uniformly over the surface of the cluster (Figure 5). In the surface structures, they are almost absent in the “bowl” region, near the surface of which the sodium cation is located, and all the molecules closest to it have both OH groups bonded. It is this kind of local fragment that can exist within the boundary shell of a nanobubble. As a result, sodium cations can be embedded in the first two superficial layers of water molecules around the gas core of a nanobubble. This should not lead to the appearance of OH groups oriented inside the nanocavity and, hence, able to locally destabilize the H-bond network at its boundary.

The hydrated sodium cation is inclined to attach an additional electron; both in the case of bulk and surface solvation, its vertical electron affinity (*VEA*) is close to 1 eV, since the value is predetermined chiefly by the configuration and, hence, the energetic state of the first solvation shell, within which the excess electron is localized (Table 2). In the case of the localization in the center of the cluster, variations in the *VEA* values are smaller. In the surface clusters, the extension of the electron localization region leads to its greater stabilization and an increase in *VEA*. The latter is possible when a sodium nucleus is located only slightly “below” the mean plane of the oxygen nuclei of the surface water molecules (as in the [(H_2_O)_51_]^+^ cluster, see Figure 5) rather than within the next (deeper) layer of water molecules, as in clusters with *n* = 38 or 43. Taking into account the lower values of the adiabatic potential of aqua complexes with the bulk solvation of a sodium cation, one can treat the value of 1 eV as a fairly good estimate of the electron affinity of hydrated sodium ions in an aqueous solution.

Thus, occupying an intermediate position between typical chaotropic (potassium cation) and kosmotropic (lithium cation) ions, (judging from the data presented above) sodium cation should introduce only a minor disturbance in the H-bond network of water. With nearly equal probability, it can be built in the bulk and superficial layers of water, including boundary spherical layers around the gas cores of nanobubbles. In the presence of excess electrons in the system, these cations can act as efficient binding sites of these electrons in the absence of stronger acceptors.

#### 2.2.2. [Cl(H_2_O)_n_]^−^ Aqua Complexes

Chloride anion, unlike sodium cation, is quite a typical chaotropic ion. Such particles are characterized by a weaker coordinating ability with respect to solvating water molecules. This is manifested in a certain orientational disordering of molecules in the first coordination sphere of the ion and their stronger inclination to form bonds with each other. As a result, aggregation of water molecules in one hemisphere of the ion is possible, which makes the surface configurations of the [Cl(H_2_O)*_n_*]^−^ aqua complexes more clearly pronounced. Unlike sodium cation, which is partially immersed in the H-bond network in the surface structures, the chloride anion can be located directly in the mean plane of the centers of masses of the neighboring water molecules. Figure 6 shows the most typical (of those obtained during optimization) structures of [Cl(H_2_O)*_n_*]^−^ clusters with the bulk and surface location of chloride anion. Even in the presence of 44 water molecules, a structural hemisphere filled with molecules can be formed, and the chloride ion is localized in the central part of its boundary layer, which is no more than 0.5-Å thick. Similar structures exist at a smaller number of water molecules as well. In this case, chloride ion forms up to five Cl^…^H bonds with the nearest molecules directly within this boundary layer. In addition, the ion forms bonds with one or two more molecules in the bulk part of the cluster. At a smaller number of water molecules (up to 35), the boundary layer of the cluster that involves the chloride ion is almost flat. At a larger number of water molecules, it can even be slightly concave, so that taking into account the more distant water molecules, we can easily imagine the inclusion of this cluster in the effective bimolecular boundary layer of a nanobubble. And in the resulting nanobubble hydration shell, the ion shall find itself directly on the surface of the gas core (which is well illustrated by the structure with *n* = 44).

In general, in the presence of 20 to 30 water molecules around a chloride ion, its coordination number equals 7 to 9, usually being larger by one in the bulk structures. With an increase in the total number of molecules in the system to 35 and then to 55, aggregates with a lower coordination number of the ion become the most stable. The number is 6–7 in the bulk structures and either 6–7 up to *n* = 40 or 5–6 at large *n* values in the surface structures.

By contrast to the sodium cation aqua complexes, for which a clear change in the energetically more favorable location of the cation from the surface to the bulk one was noticed at a certain (sufficiently large) number of water molecules, the hydration behavior of the chloride anion is more complex. The *H_rel_*/*n* values of the [Cl(H_2_O)*_n_*]^−^ bulky clusters are close to 17.7 kcal/mol (vary from 17.64 to 17.75 kcal/mol), while for the surface aqua complexes, they fall in a broader range, from 17.5 to 18.0 kcal/mol. For example, for the [Cl(H_2_O)_47_]^−^ cluster, the surface structure is more energetically favorable (*H_rel_* values differ by 10.4 kcal/mol). However, for the [Cl(H_2_O)_55_]^−^ cluster, the bulk structure is preferable (with an even greater difference in *H_rel_* values, about 13.0 kcal/mol). A reason for that is the drastic difference in the coordination of the outer water molecules of the cluster and the overall “compactness” of the H-bond network. For example, at *n* = 55 in the bulk structure, only four OH groups are not involved in the H-bond network, and the configuration of the complex is close to spherical (Figure 6), while in the surface structure, twice as many OH groups are not H-bonded, and the H bond network is significantly less optimal, because many outer molecules belong to a more extended hydration sphere, which is incomplete. There is an additional aspect that is important in the context of the possible inclusion of ions in the boundary molecular layers of nanobubbles. Note that half of the eight free (H-unbound) OH groups are located quite close to the chloride anion. This should cause the local instability of such a boundary segment. At the same time, in a cluster with *n* = 47, the number of free OH groups in the surface structure is one less (5 vs. 6), and they are oriented away from the ion. Hence, these groups can easily be involved in H-bonds with water molecules of the next neighboring layers without destabilizing the boundary layer of a nanobubble.

Vertical ionization of the chloride anion aqua complexes requires a noticeable energy (about 6.5 eV; see Table 3). This is due to both the high ionization potential of the individual chloride ion (3.63 eV) and the stabilizing role of the surrounding water molecules directly involved in the localization of the corresponding electron density: the effective charge on chlorine is −0.63–−0.70 a.u., and the total negative unit charge is concentrated within its first solvation shell. In addition, the more water molecules provide their OH groups for the formation of Cl^…^H-O coordination bonds, i.e., the higher the coordination number of the ion, the stronger the electron density redistribution between the ion and water molecules.

Thus, chloride anions should easily be incorporated in the boundary layers of water molecules around the gas cores of nanobubbles and localized more closely to this core compared to bicarbonate ions [1], since the stable binding of the latter requires at least three molecular layers of water. Chloride ions can be located directly within the first superficial layer of molecules with a thickness of about 0.5 Å and form bonds with the nearest molecules. In this case, they can lock OH groups, which are not involved in the formation of the H-bond network, thus additionally stabilizing the boundary layer of a nanobubble. The intrinsic charge of these ions turns out to be very efficiently and steadily localized within their first solvation shells.

#### 2.2.3. Excess Electron Localization in Water Clusters: [(H_2_O)_n_]^−^

As we already noted, hydrated electrons are localized in D-type Bjerrum defects of the H-bond network, being located in the center of the region where the OH groups of several water molecules are directed. The number of such groups can vary from two to six, and, as was shown in our earlier work [28], analysis of even small clusters of water molecules enables one to estimate the hydration energy of an electron. The latter is possible because the character of the excess electron distribution in water clusters does not change substantially with an increase in the number of molecules in the second and next solvation shells. This means that qualitative differences in the state of an excess electron depending on the structure of its first hydration shell can be clarified based on the analysis of relatively small clusters, which allows for the corresponding variations in the mutual arrangement of molecules. In this study, as such a cluster, we have taken (H_2_O)_27_, which enables us to compare the energy characteristics of the structures, in which the number of water molecules directly involved in the excess electron localization varies from three to six. The latter values correspond to the minimum and maximum size of the first hydration shell of an electron in the bulk structures.

Figure 7 (top) shows the optimized structures of [(H_2_O)_27_]^−^ anion. We found stable configurations in which OH groups of three, four, and six water molecules participate in the excess electron localization (i.e., the formal coordination number, *CN*, of the electron is 3, 4, or 6). Obviously, the greater the number of molecules involved in the excess charge localization, the more noticeably the corresponding cluster fragment differs structurally from what is typical of the H-bond network of water (cf. structures shown in the top and bottom rows of the figure). Hence, the greater the structure reorganization energy, *E_reorg_*. This means that a large *CN* value of the hydrated electron is possible only when the excess electron appears in a region of a substantially perturbed structure (with numerous defects in the H-bond network) of water. It is noteworthy that the energies of the water cluster anion with different *CN* values do not differ very strongly (see Table 2). The structure with the lowest energy is the one where the excess electron is localized in the surface layer of the cluster (at *CN* = 4). At the same time, for all variants of the electron localization, the vertical electron detachment energy (*VDE*) is quite high, increasing with the growth in the number of molecules in the first hydration sphere (Table 2). At the same time, the adiabatically stable electron localization is possible only at *CN* = 6, when a defective H-bond network of water molecules is formed. This is reflected by the adiabatic detachment energy *ADE* = *E*((H_2_O)*_n_*)) − *E*([(H_2_O)*_n_*]^−^), where *E*((H_2_O)*_n_*) is the energy of the cluster after its relaxation caused by the removal of an electron (Table 4).

Thus, when the perturbation of the H-bond network of water is sufficient for the local rupture of H-bonds and restricted rotations of individual molecules, one can expect that the bulk localization of excess electrons by water molecules, the local coordination of which is close to octahedral, should be energetically favorable and optimal. Note that the corresponding perturbation should not be too large, because the H-bond network around a relatively small region of the excess electron localization should have a small excess (thermal) energy. Otherwise, a perturbation rather than stabilization of the region becomes possible. At a stronger perturbation of the structure, the excess electron localization of the surface kind should be more probable at an interface of the bonded molecular layers and the appeared defect, which may have dimensions of a nanometer to hundreds of nanometers. In this case, the excess electron is localized by a molecular hemisphere composed of three to four molecules.

Two more examples of the surface and subsurface kinds of excess electron localization are shown in Figure 8. In the structure with the surface localization (Figure 8a), an electron is captured by a molecular hemisphere on the surface of a larger defect. A concave region of the bond network has emerged on the convex surface of the H-bonded cluster. Within this region, OH groups of the molecules uninvolved in H-bonds localize the excess electron. This kind of fragment can appear on a slightly concave or slightly convex surface. In another structure (with the subsurface localization, Figure 8b), such hemisphere is located inside a large hollow cluster, which mimics the localization of an excess electron inside a spherical boundary of a nanobubble.

In the former case, the *VDE* of an electron is 1.78 eV, which, on the one hand, almost exactly corresponds to the maximum of the experimental absorption spectrum of hydrated electrons, and, on the other hand, fits very well into the dependence of the *VDE* of water anions with the surface charge localization, which was proposed by us previously: *VDE* = 2.93 − 4.42 *n*^−1/3^. Note that this dependence was constructed on the basis of data for cluster anions composed of no more than 16 molecules [28]. This means that this functional dependence, which was deduced from the analysis (expansion in spherical harmonics) of the differential electron density of neutral and negatively charged clusters at an assumption of its locality, is not only physically correct but also enables one to estimate the unknown characteristic for the systems of arbitrary molecular size. Note also that when the number of molecules is no larger than ~110, the *VDE* value is less than 2 eV, which corresponds to a wavelength of 600 nm, at which the absorption of radiation by the samples was recorded in our experiments. Hence, the existence of defects in the H-bond network, which restrict the localization regions of the excess electrons, should lead to a lower energy of their binding and the possibility of their transition to the delocalized rather than the first excited (p-type) state at the same energies (wavelengths) of electromagnetic radiation.

In the case of subsurface localization of the excess electron, the electron detachment energy is even lower, 1.21 eV, due to the weaker distortion of the water cluster and the simultaneously smaller spatial extension of the electron localization region. In fact, the excess electron localized by three OH groups of water molecules is in a state close to that identified above in the [(H_2_O)_27_]^−^ cluster (Figure 7, structure with *CN* = 3). The increase in its vertical detachment energy with an increase in the number of molecules in the cluster from 27 to 60 is due to the expansion of the cavity inside the water cluster, i.e., weaker spatial restrictions. Thus, at the surface or subsurface localization of the excess electron, the weaker the distortion of the cluster structure in comparison with a neutral molecular aggregate and the smaller the spatial extension of the localization region, the lower the binding energy of the electron. At the same time, in all cases, the absorption of energy of 2 eV should cause a detachment rather than an excitation of the electron. The above estimates correspond to the transition of an electron to a free state in the gas phase.

Note also that significantly higher *VDE* values (about 2.7 eV) correspond to the electron localization by an octahedron of OH groups of six water molecules surrounded by at least twenty more molecules. This situation can take place in the bulk water phase after structural relaxation caused by the appearance of an excess electron. Under these conditions, electromagnetic radiation in the range we use should lead to the electronic excitation of the corresponding fragment of the H-bond network of water. At the same time, under the experimental conditions when we can expect the existence of numerous nanobubble defects, on the surfaces of which secondary electrons can be localized, the energy supply of 2 eV should be sufficient for the detachment of an electron from the solvating molecules.

## 3. Discussion

The samples under investigation contain bicarbonate ions formed as a result of the hydration of carbon dioxide dissolved in water, see Equation (1). Taking into account that the solubility of CO_2_ in water under normal conditions (298 K, 1 atm) is 0.48 mg/L, the concentration of bicarbonate ions in the samples that came to equilibrium with the atmospheric air is approximately 10^19^ ion/L. In the absence of a vibrational treatment, these ions are solvated in the bulk phase, being prone to form at least five H-bonds with the nearest water molecules [1]. Vigorous shaking should promote partial accumulation of bicarbonate ions in the spherical shells of nanobubbles, where these ions are separated from the internal gas core of the nanobubble by one or two layers of water molecules [1]. Water molecules closest to a bicarbonate ion are oriented so that their OH groups are directed almost radially away from this ion. This can potentially contribute to the formation of regions at some distance from bicarbonate ions, within which excess electrons can be localized.

In the presence of dissolved sodium chloride, the solubility of carbon dioxide in water decreases by 4.5%, which seems to be an insignificant change. However, due to the greater inclination of chloride ions to be localized directly at the water-air interface (see above), these ions should partially replace bicarbonate ions in the spherical shells of gas nanobubbles by displacing them into the bulk water phase.

Hydrated electrons, as follows from the above results, can easily be formed in the bulk water, especially when the water sample is sufficiently disturbed (for example, under the effect of a high-energy electron impact). In this case, the surface localization of an excess electron in the vicinity of a sufficiently large defect in the H-bond network seems energetically preferable, since it requires a less noticeable reorganization of molecules (see Table 4). This means that in the presence of nanobubbles, hydrated electrons should be more stable in their shells. However, the inclusion of an excess electron in such a shell requires overcoming the energy barrier caused by the repulsive electrostatic potential created by chloride and bicarbonate ions. On the one hand, given that secondary electrons have an energy of about 10 eV, one can expect them to overcome easily the corresponding energy barrier. At the same time, the preferential localization regions of excess electrons are the same defects in the H-bond network of superficial water molecules as those localizing chloride ions (compare the surface structures of [Cl(H_2_O)*_n_*]^−^ anions in Figure 6 and the structure of [(H_2_O)_27_]^−^ anion with *CN* = 4 in Figure 7). Therefore, the number of secondary electrons, whose state can be stabilized in the boundary layers of nanobubbles, should be smaller in the presence of chloride ions in the sample.

An additional factor is that the increase in the number of anions localized in the boundary layer of a nanobubble (chloride ions together with bicarbonate ions) should be accompanied by a proportional (not necessarily equal) increase in the number of counterions (sodium ions) localized in relative proximity to the boundaries of nanobubbles in the diffuse layer. An increase in their concentration in the vicinity of nanobubbles should produce an electrostatic trap on the path of diffusing excess electrons, slowing them down near the boundary of the diffuse layer. At the same time, sodium cations themselves should act as very efficient local traps. As follows from the data presented above, the vertical electron affinity of sodium ion aqua complexes is always positive and close to 1 eV, whereas that of defect-free fragments of the H-bond network of water molecules is practically zero. Even, in the presence of D-type Bjerrum defects it amounts to only several kcal/mol (see above). It means that in the presence of NaCl dissolved in water, secondary electrons should initially be captured predominantly by [Na(H_2_O)*_n_*]^+^ fragments. However, if a specimen is irradiated with electromagnetic waves of at least 1 eV energy, most of the localized electrons will be removed from the corresponding solvation shells and become hydrated.

Thus, in general, in the presence of NaCl ions, the region of the bicarbonate localization should become more distant from the gas core of nanobubbles; and it is chloride ions and secondary electrons that should be localized nearly at the shell/core interface. Outside the boundary hydration shell, sodium and hydronium ions should reside, their relative positions with respect to the core correlating to their mobilities in water. All these conclusions are schematically summed up in Figure 9, which supplements the basic scheme shown in Figure 4.

The number of nanobubbles in different samples of water and aqueous NaCl solutions can be estimated on the basis of the following data. According to [7], in a cell with initially degassed water open to the atmosphere, the volume number density of nanobubbles, *n_b_*, increases according to diffusion kinetics. Under the equilibrium conditions in the presence of gas components of air dissolved in water (with a content of about 10^17^ cm^−3^) and in the presence of NaCl salt ions with a concentration of 10^−4^–10^−3^ M, *n_b_* falls in a range of 10^6^–10^7^ cm^−3^. Vibrational treatment of samples contributes to an increase in *n_b_* to 10^9^–10^10^ cm^−3^, i.e., by three orders of magnitude [1].

Taking into account the models discussed above, we can expect that electromagnetic radiation with a wavelength of 600 nm should lead to the excitation of localized hydrated electrons in the bulk water phase and to the removal of hydrated electrons from the nanometer-sized regions of their surface or subsurface localization in the boundary shells of nanobubbles (in the absence of a noticeable amount of dissolved NaCl). The relative number of electrons in such regions should decrease when sodium chloride is dissolved in water and increase after vibrational treatment of the samples, which generates a large number of nanobubbles with diameters of 50 to 150 nm.

The simplest kinetic scheme (*scheme 1*) that can be used to estimate the number of hydrated electrons in the absence of dissolved salt in a water sample is as follows:*e*_HE_ → *m e*_aq_ (*k*_1_),*e*_aq_ → *e*_exc/deloc_ (*k*_2_).
where *e*_HE_ is the primary high-energy electron, *e*_aq_ is the hydrated secondary electron, and *e*_exc/deloc_ is the electron excited or knocked out from the localization region by the momentum of light quantum. The *m* values can be estimated based on the experimentally determined yield of hydrated electrons, which is approximately equal to 10^−7^ mol/J [29]. The apparent rate constants of the first (*k*_1_) and second (*k*_2_) stages (as effectively monomolecular processes) characterize a very fast process leading to an increase in the number of secondary hydrated electrons (with the rate *k*_1*eff*_ = *mk*_1_) and a slower process predetermined by the radiation absorption probability, respectively. In general, the recorded signal intensity at a wavelength of 600 nm is determined by the number of hydrated electrons $n_{e_{aq}}$ and their lifetime. The latter can be estimated as 1/*k*_2_, while the former, for the above scheme at an assumption of the absence of *e*_aq_ in the system before the high-energy electron impact, depends on time as follows (Equation (2)):(2)$$n_{e_{aq}}\left(t\right)=n_{e_{HE}}\frac{mk_{1}}{k_{2}-k_{1}}\left(e^{-k_{1}t}-e^{-k_{2}t}\right).$$

The signal profile (see Figure 1) can be approximated by a function $A=\varepsilon n_{e_{aq}}\left(t\right)$, where ε is the extinction coefficient, which for hydrated electrons at a wavelength of 600 nm, according to [29], can be set to 16,380 M^−1^cm^−1^. The *m* values under the experimental conditions are ~4·10^5^. Then, for water samples in the absence of vibrational treatment, we come to the following estimates: *k*_1_ = 0.76 μs^−1^ and *k*_2_ = 1.00 μs^−1^.

For the NaCl solution, a similar approximating function gives poor results due to a noticeably slower increase in the absorbance and its smaller value at the signal maximum at a similar character of the absorbance decay. This is a consequence of the above-mentioned role of sodium cations in the binding of secondary electrons in the solution. Taking into account that the salt concentration is 10 mg/L (i.e., 0.17 mM, which corresponds to the presence of ~10^20^ sodium ions in a liter of solution, which exceeds the number of generated secondary electrons by more than one exponent), one can expect that almost all the secondary electrons are initially captured by the hydration shells of sodium cations. However, under the experimental conditions, when irradiating the sample with a xenon lamp with a broad spectrum of radiation, most of these localized electrons will be removed from the corresponding solvation shells. This will lead to the appearance of hydrated electrons. In this case, the kinetic scheme of the process (*scheme 2*) can be written as follows:*e*_HE_ → *m e*_s(Na)_ (*k*_1_),*e*_s(*Na*)_ → *e*_aq_ (*k*_2_),*e*_aq_ → *e*_exc/deloc_ (*k*_3_).

It implies an inevitably successive change in the state of secondary electrons. An alternative version allows for the possibility of competing processes of the electron capturing by the solvation shells of sodium cations and H-bond network defects of water with the formation of hydrated electrons (*scheme 3*):*e*_HE_ → *m*_1_ *e*_aq_ (*k*_1_),*e*_HE_ → *m*_2_ *e*_s(Na)_ (*k*_2_),*e*_s(*Na*)_ → *e*_aq_ (*k*_3_),*e*_aq_ → *e*_exc/deloc_ (*k*_4_).

The change in the concentration of hydrated electrons with time in the case of *scheme 2* is given by a function of Equation (3):(3)$$n_{e_{aq}}\left(t\right)=n_{e_{HE}}mk_{1}k_{2}\left(\frac{e^{-k_{3}t}}{k_{3}^{2}+k_{1}k_{2}-(k_{1}+k_{2})k_{3}}+\frac{e^{-k_{2}t}}{k_{2}^{2}-k_{1}k_{2}+(k_{1}-k_{2})k_{3}}+\frac{e^{-k_{1}t}}{k_{1}^{2}-k_{1}k_{2}+(k_{2}-k_{1})k_{3}}\right).$$

In the case of *scheme 3*, the time dependence of the concentration of hydrated electrons is as follows (Equation (4)):(4)$$n_{e_{aq}}\left(t\right)=n_{e_{HE}}\left(\begin{array}{l} \frac{-m_{1}k_{1}k_{4}+\left(m_{1}k_{1}+m_{2}k_{2}\right)k_{3}}{k_{4}^{2}+\left(k_{1}+k_{2}\right)k_{3}-\left(k_{1}+k_{2}+k_{3}\right)k_{4}}e^{-k_{4}t}+\frac{m_{2}k_{2}k_{3}}{k_{3}^{2}-\left(k_{1}+k_{2}\right)k_{3}-\left(k_{3}-k_{2}-k_{1}\right)k_{4}}e^{-k_{2}t}+\\ +\frac{-m_{1}k_{1}\left(k_{2}+k_{1}\right)+\left(m_{1}k_{1}+m_{2}k_{2}\right)k_{3}}{{\left(k_{1}+k_{2}\right)}^{2}+\left(k_{3}-k_{2}-k_{1}\right)k_{4}-\left(k_{1}+k_{2}\right)k_{3}}e^{-\left(k_{1}+k_{2}\right)t} \end{array}\right).$$

The use of such functions gives an approximation of the experimental signal of practically the same quality (Figure 10). In this case, parameters of the kinetic schemes are as follows: *k*_1_ = 0.8, *k*_2_ = 1.0, *k*_3_ = 1.4 μs^−1^ in the case of *scheme 2*; and *k*_1_ = 0.08, *k*_2_ = 0.75, *k*_3_ = 1.0, *k*_4_ = 1.4 μs^−1^ at the *m*_2_/*m*_1_ ratio of 115 in the case of *scheme 3*. The very small value of *k*_1_ together with the difference between the *m*_2_ and *m*_1_ values by more than two exponents and the proximity of the residual constants mean that the primary capture of electrons occurs almost exclusively at the solvation shells of sodium cations. Hence, the simpler (sequential) *scheme 2* represents the processes in a NaCl solution quite reliably. In this case, the integral signal intensity is ~0.19 rel. units compared to the value of ~0.25 rel. units for pure water, i.e., is 25% less, which correlates with the peak absorbances (where the difference is 28%, see Figure 2). Thus, in the absence of vibrational treatment, the experimental results correlate well with theoretical estimates and predictions.

In the case of a vigorous vibrational treatment of ultrapure water or aqueous NaCl solution during their repeated dilution, the spectral signal profile and its parameters are almost independent of whether the initial sample contained salt or not. At the same time, the profile itself differs from those for ultrapure water and NaCl salt solution that were not subjected to vibrational treatment. The signal decays noticeably more slowly, and the average asymptotic value remains nonzero for more than additional 20 μs, which reflects the longer presence of particles in the system that are capable of absorbing radiation at a wavelength of 600 nm. This character of the absorbance changes cannot be described without an assumption about the reversibility of the last stage in the kinetic schemes used above. This is possible if only we assume that a noticeable part of the signal at 600 nm is due to the energy absorption, which causes the removal of an electron from the region of its surface or subsurface localization. In accordance with the above simulation data, this is quite possible. Then, depending on the type of the environment around the electron localization region, its dynamics should be different. In the case of relatively small defects in the H-bond network, it can be assumed that the removed electron ends up in the bulk phase of the liquid sample, where it is almost inevitably (on the femtosecond scale) captured by some electron-deficient particle. In the samples subjected to vigorous vibrational treatment, there are numerous not only nanometer-sized defects but also fairly large nanobubbles. Accordingly, a significant portion of the secondary electrons can be localized in the boundary shells of such nanobubbles at a side of the gas core. When detached (due to the absorption of 2-eV energy), these electrons should find themselves in the gas core region. Their kinetic energy will be sufficient to cover the distance to the other (opposite) segment of the gas nanobubble in sub-picosecond time, where their repeated localization is possible. The localization should almost exclusively be superficial, which requires the minimal rearrangement of molecules. Such electrons can be re-transferred from a localized to a delocalized (free) state upon absorption of 2-eV energy, which should provide an additional contribution to the spectral signal at later times, which should cause a slower decay of the recorded signal.

Thus, in the presence of nanobubbles in the system, the basic kinetic scheme requires the following modification (*scheme 4*):*e*_HE_ → *m e*_aq_ (*k*_1_),*e*_aq_ → *e*_exc/deloc/free_ (*k*_2_),*e*_free_ → *e*_aq_ (*k_−_*_2_).

Here, we took into account the reversibility of the last stage, which corresponds to the appearance of an electron, which is not delocalized in the bulk condensed phase but is rather free in the gas phase of the nanobubble. This electron can again be captured by another surface segment of the boundary hydration shell of the same nanobubble. For such a scheme, the time dependence of the number of hydrated electrons (which determine the recorded spectral response) looks as follows (Equation (5)):(5)$$n_{e_{aq}}\left(t\right)=n_{e_{HE}}m\left(\frac{-k_{1}k_{2}e^{-\left(k_{2}+k_{-2}\right)t}}{k_{2}^{2}+k_{-2}^{2}-k_{1}k_{2}+\left(2k_{2}-k_{1}\right)k_{-2}}+\frac{\left(k_{1}-k_{-2}\right)e^{-k_{1}t}}{k_{2}+k_{-2}-k_{1}}+\frac{k_{-2}}{k_{2}+k_{-2}}\right).$$

Parameters of the approximating function are as follows: *k*_1_ = 0.4, *k*_2_ = 0.4, and *k*_−2_ = 0.004 μs^−1^. Two points should be noted. (i) The constants of both the first and second stages are approximately half of those in the case of samples that were not subjected to vibrational treatment. This confirms a different character of the localization of secondary electrons in the system and a different character of the process initiated by the absorption of electromagnetic radiation. (ii) The rate constant of the reverse stage (electron rehydration), which ensures a slower decrease in the absorbance and its non-zero value over a sufficiently long time interval, is very small: lower by two exponents than that of the forward reaction stage. This means that the fraction of recaptured and then excited electrons in the solution is very small. However, in the absence of an assumption about their presence in the system, the profile of the experimental spectral signal cannot be described. Moreover, it is impossible to explain the three times greater integral value of this signal: ~0.79 rel. units (compared to 0.25 rel. units for water, which was not vibrationally treated; see above).

## 4. Materials and Methods

### 4.1. Sample Preparation

All samples were prepared using ultrapure type 1 water (with a specific resistance of 18 MΩ × cm at 25 °C immediately upon purification) produced using a Milli-Q purification system (Millipore, Merck KGaA, Darmstadt, Germany).

At the first step, NaCl (Sigma Aldrich, St. Louis, MO, USA) aqueous solution with a concentration of 10 mg/L was prepared. Then it was subjected to a series of successive hundredfold dilutions, which were accompanied by controlled vigorous mechanical shaking at each stage of dilution. For this purpose, a tube that contains a sample was intensively shaken manually with a fixed frequency of about 4 Hz (21 strokes in about 4.8 s with a force of 4 N (kg m/s^2^)). The frequency and intensity of shaking were monitored with a custom-made Dynamizer device using a Tenzometry Unit online monitoring software ver. 3.1.1 developed based on the Laboratory Virtual Instrumentation Engineering Workbench (LabVIEW 2019, Version 1.4.15.5.). The total number of hundredfold dilutions was no less than 12, with a water-ethanol mixture used at intermediate stages and ultrapure water, at two final stages. A theoretical decrease in the concentration compared to the NaCl stock solution was at least 10^24^-fold. Samples were prepared in 40 mL vials (Glastechnik Grafenroda, Geratal, Germany) and then poured into 500 mL glass jars (Simax^®^, Praha, Czech Republic). The samples were stored in the jars at room temperature protected from direct daylight; the lids of jars being tightly closed.

The liquid samples used in the experiments are listed below:Water—intact liquid not subjected to shaking (reference sample);«HD VT NaCl»—vibrationally treated NaCl solution subjected to repeated (high) dilution;«HD VT water»—vibrationally treated water subjected to repeated (high) dilution similarly to «HD VT NaCl» samples.

### 4.2. Photometric Determination of the Content of Radiation-Absorbing Particles

Irradiation of liquid samples with a high-energy electron beam was carried out with the use of the linear electron accelerator. This accelerator generates electron pulses of 1.5 µs duration, 150 mA current (which corresponds to approximately N = 1.5 × 10^12^ electrons in a pulse), and 6 MeV energy. The electron energy absorption in our experiments corresponded to 5 × 10^3^ Gy or 50 J/kg. When a liquid sample is irradiated with an electron beam, numerous side processes that consume the primary electron energy take place. These are heating and radiolysis of water accompanied by the generation of excited OH• radicals (2.65), hydrated electrons, *e_aq_*^−^, (2.65), excited hydrogen atoms H• (0.60), molecular hydrogen (0.45), and hydrogen peroxide (0.75); see, e.g., [30]. Here the numbers in parentheses show the yields of products in molecules (radicals) per 100 eV of absorbed energy in water. The experimental setup is shown in Figure 11.

An optical flow cell filled with a liquid sample (2) with a volume of 100 mL and an optical path of 10 mm was irradiated with an electron beam generated in the linear accelerator (1) and concurrently illuminated with a broadband white light source (3) (150 W xenon lamp with a spectral range of 250–800 nm). The radiation of the lamp (3) was collimated into the cell (2) with the use of a system of lenses and mirrors (4). After passing through the cell (2), the lamp radiation (3) was collimated with a folding mirror (5) to a monochromator (6) equipped with a diffraction grating. The radiation at a wavelength of 600 nm separated with the monochromator was supplied to the entrance slit of the photomultiplier tube (7). The signal from the photomultiplier was sent to the ADC (8) and processed by a computer (9). In this way, it was possible to measure the transmittance of the liquid samples at a wavelength of 600 nm with high accuracy. For the subsequent analysis, the transmittance was converted to the absorbance.

### 4.3. Experimental Data Analysis

The data analysis and visualization were carried out with the use of R (version 4.0.2, R Foundation for Statistical Computing, Vienna, Austria) and Microsoft Excel (version 2016, build 16.0.2566.1000). The results are presented as the arithmetic mean ± standard deviation (SD). The Shapiro–Wilk test was employed to evaluate the normality of the data distribution, while the Bartlett test was used to assess the variance homogeneity. Group comparisons were based on the Student’s *t*-test or on Mann–Whitney U test. Statistical significance was defined as *p* < 0.05.

### 4.4. Quantum Chemical Simulations of the State of Secondary Electrons and Chlorine and Sodium Ions in Water

To model the state of secondary (excess) electrons, sodium cations and chloride anions in water near or inside the boundary layer around the gas core of a nanobubble (see [1]), quantum chemical calculations of model clusters of the following composition were carried out: [(H_2_O)_n_]^−^, [Na(H_2_O)*_n_*]^+^ and [Cl(H_2_O)*_n_*]^−^ with the number of molecules *n* ≤ 60. For bicarbonate ion, various hydration structures were considered in our previous study [1]. When constructing the initial approximations for the ionic clusters, various variants were considered, namely, (i) a gradual increase in the number of water molecules around the ion, either uniformly on all sides of the ion or predominantly within a part of the solvation sphere of the ion, and (ii) a random removal of some water molecules from a larger cluster to create significant potential energy gradients, which promote a substantial reorganization of the residual water molecules. When designing the initial structures of water clusters, which localize the excess electron, the tendency of the electron to be localized in D-type Bjerrum defects, i.e., in the areas where OH groups of water molecules face each other with their hydrogen atoms, was taken into account.

All structures were optimized, and their correspondence to the adiabatic potential minima was confirmed by normal coordinate analysis. For systems of the same composition but different configurations, the relative values of the adiabatic potential with respect to the deepest minimum (*E_rel_*) were found. Thermal increments in the energy were estimated within the framework of statistical thermodynamics by taking into account the vibrational contributions solely to the enthalpy of the systems under normal conditions (298 K, 1 atm) at an assumption about the validity of the Gibbs canonical ensemble statistics (*H_vib_*). For characterizing the stability of cluster systems, the relative enthalpies *H_rel_ = E_rel_ + H_vib_* were normalized to the n, number of water molecules in the cluster (*H_rel_/n*).

The vertical detachment energy (VDE), which characterizes a practically instantaneous removal of an electron from the cluster when the configuration of the nuclear subsystem has no time to relax, was estimated for chloride anion aqua complexes and water anions as follows:*VDE_Cl_aq_ = E*′(Cl(H_2_O)*_n_*) − *E*([Cl(H_2_O)*_n_*]^−^),*VDE_water_ = E*′((H_2_O)*_n_*) − *E*([(H_2_O)*_n_*]^−^).

Both *E*′(Cl(H_2_O)*_n_*) and *E*([Cl(H_2_O)*_n_*]^−^) or *E*′((H_2_O)*_n_*) and *E*([(H_2_O)*_n_*]^−^) energies were calculated at the optimal configurations of the corresponding anion ([Cl(H_2_O)*_n_*]^−^ or [(H_2_O)*_n_*]^−^). The prime indicates that the structure of the corresponding particle is not optimal.

The vertical electron affinity (VEA) of water clusters or sodium cation aqua complexes were estimated in the same way, namely, as the differences in the energies of the resulting particle (water cluster anion or sodium atom aqua complex) and the initial particle, which have the same configuration corresponding to the minimum of the adiabatic potential of the initial particle:*VEA_water_* = *E*′([(H_2_O)*_n_*]^−^) − *E*((H_2_O)*_n_*),*VEA_Na_aq_* = *E*′([Na(H_2_O)*_n_*]) − *E*([Na(H_2_O)*_n_*]^+^).

As earlier, the prime indicates that the structure of the corresponding particle is not optimal.

The structure reorganization energy upon removal of an excess electron was estimated as the difference between the energy of the neutral cluster at the optimal configuration of the anion (*E*′((H_2_O)*_n_*)) and the energy of the structure that forms upon relaxation caused by the electron removal (*E*((H_2_O)*_n_*)):*E_reorg_* = *E*′((H_2_O)*_n_*) − *E*((H_2_O)*_n_*).

The structures were optimized at the density functional level using the B3LYP hybrid exchange-correlation functional [31,32,33,34], which accurately predicts the character of the electron density distribution in water clusters, including those with a nonzero charge caused by the presence of either an excess electron or foreign sodium or chloride ions. To approximate the single-electron functions, a two-exponential Gaussian basis set supplemented with polarization functions on all nuclei 6-31G(d,p) (6-31G(d) in the case of O, Na, and Cl and 6-31G(p) in the case of H) [35,36,37] was used. This basis is flexible enough to describe clusters that involve more than two dozen molecules and, at the same time, relatively compact to eliminate the possibility of a linear dependence of the functions. In the analysis, we used our previously obtained data on the structure and dynamics of sodium cation aqua complexes [27].

To determine the excitation energies of cluster systems, the TDDFT method was used, and the energies of the 20 lowest electronic states of clusters were found. Nonempirical calculations were carried out with the use of quantum chemical package Firefly 8.2 [38], which is partly based on the GAMESS (US) software code [39], and the Orca 6.0 package [40,41,42,43]. The Chemcraft graphical package, ver.1.8 [44] was used for the analysis and visualization of the results.

## 5. Conclusions

Summarizing the results and estimates obtained in this study, we can state the following. Measuring the absorbance of water and aqueous solutions at 600 nm upon the high-energy electron impact reveals differences in the states of samples depending on their preliminary treatment. Upon the repeated vibrational treatment (vigorous shaking) of the samples, both the maximum and integral absorbance were found to increase substantially within the 10 to 20 μs time interval after the onset of the electron impact. Insofar as the absorbance in this range is determined by the excitation of (secondary) hydrated electrons, both the amount and apparent lifetime (the duration of the time interval when particles are present in the environment and can absorb radiation) of these electrons should increase upon shaking. Taking into account that repeated vibrational treatment promotes generation of nanobubbles in the samples, it is likely that these nanobubbles should play the key role in the experimentally discovered trends. To check and prove the hypotheses, nonempirical simulations of diverse ionic species (aqua complexes of excess electrons and foreign ions) were carried out that formed grounds for the following conclusions.

One of the probable kinds of secondary electron localization in aqueous samples is the surface or subsurface one. In this situation, the photon energy of 2 eV is sufficient not only to excite a localized electron but also to cause its transition to a delocalized state in the bulk liquid. The surface and subsurface localization of excess electrons seems to be still more likely within the boundary hydration shells of gas nanobubbles. Here, a transition of hydrated electrons to a free state in the gas phase of nanobubbles can be followed by their subsequent recapturing in another segment of the boundary hydration shell of the nanobubble. This explains the data that, at first glance, do not agree with typical regularities, namely, a slower decay of the absorbance in a non-homogeneous sample. The formally longer lifetime of the light-absorbing particles, which may be supposed based on the slower signal decay, is actually the longer presence of such particles in the sample due to their reappearance upon the transient transfer to the gas phase inside nanobubbles. The corresponding kinetic schemes provided high-quality approximations of the actual profiles of the spectral signals and enabled us to estimate the rate constants of different stages, including the secondary electron generation, excitation, delocalization, and recapturing.

On the other hand, the corresponding spectral data can be considered as additional independent confirmation of the presence of a large number of nanobubbles in vibrationally treated water samples, the dynamics of which should also be manifested in other properties of such samples.

## Figures and Tables

**Figure 1 ijms-26-08909-f001:**
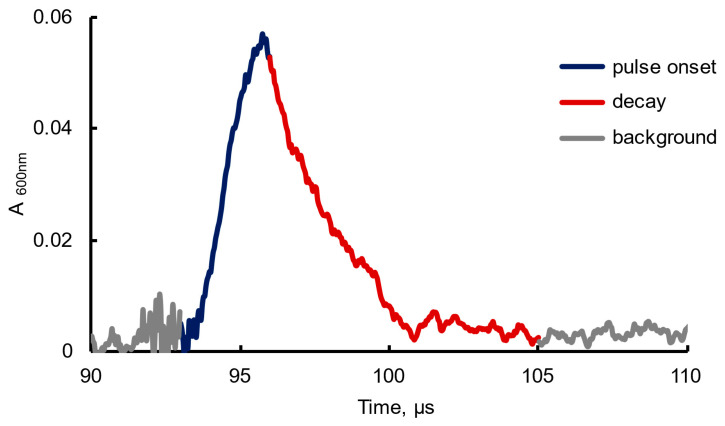
A typical time dependence of the absorbance of an aqueous sample of NaCl solution (10 mg/L) at a wavelength of 600 nm upon the high-energy electron impact (6 MeV and a 1.5-µs pulse). The pulse onset of the electron impact coincides with the beginning of the ascending segment of the absorbance peak (shown in deep blue).

**Figure 2 ijms-26-08909-f002:**
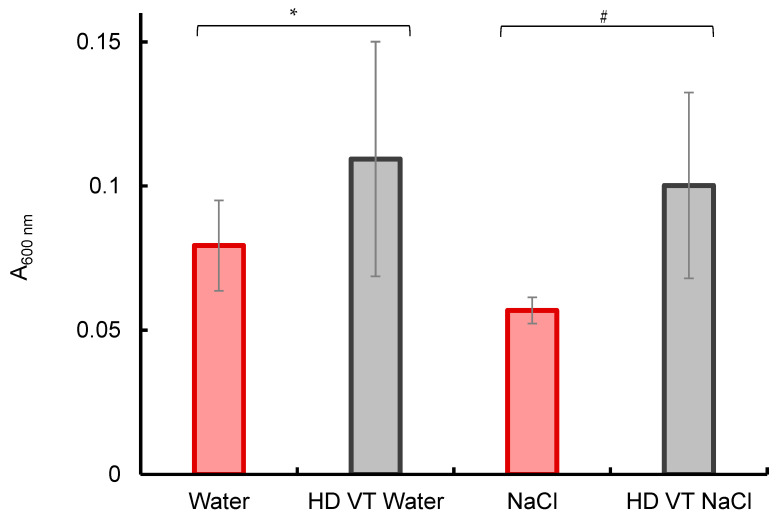
The peak absorbance at the wavelength of 600 nm of the reference sample (Water), water subjected to intensive vibrational treatment (VT) during repeated 100-fold (high) dilution (HD) steps (HD VT Water), aqueous NaCl solution with a concentration of 10 mg/L (NaCl), and aqueous NaCl solution subjected to successive dilutions accompanied by vigorous vibrational treatment (HD VT NaCl). Measurements were carried out during and after the high-energy electron impact (6 MeV, a 1.5-µs pulse). The data are shown as arithmetic means ± SD for 6 experiments. * *p* = 0.06; # *p* < 0.001.

**Figure 3 ijms-26-08909-f003:**
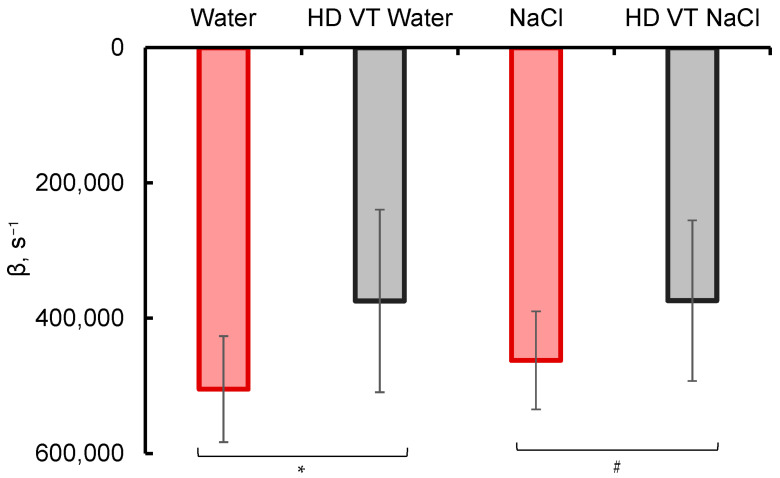
The exponential factor β, which characterizes the absorbance decay at a wavelength of 600 nm after reaching the peak value, for the reference sample (Water), water subjected to intensive vibrational treatment (VT) during successive 100-fold (high) dilution (HD) steps (HD VT water), aqueous NaCl solutions with a concentration of 10 mg/L (NaCl), and aqueous NaCl solutions subjected to intensive vibrational treatment during repeated 100-fold dilutions (HD VT NaCl). Measurements were carried out during and after the high-energy electron impact (6 MeV, a 1.5-µs pulse). The data are presented as arithmetic means ± SD for 6 experiments. * *p* = 0.02; # *p* = 0.06.

**Figure 4 ijms-26-08909-f004:**
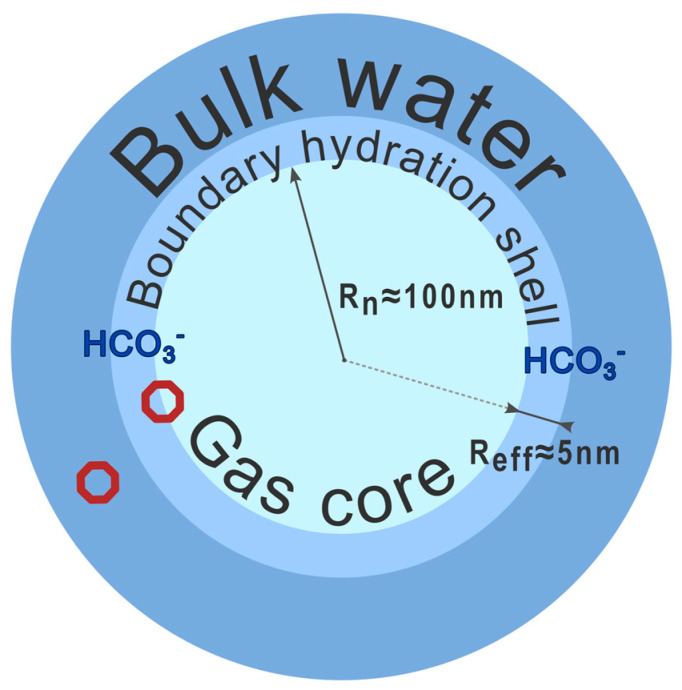
Schematic illustration of a nanobubble in bulk water, where the regions of probable localization of bicarbonate ions are indicated along with the domains (shown as octagons), to which model clusters belong. Rn, the radius of a gas core; Reff, the effective radius of a boundary hydration shell.

**Figure 5 ijms-26-08909-f005:**
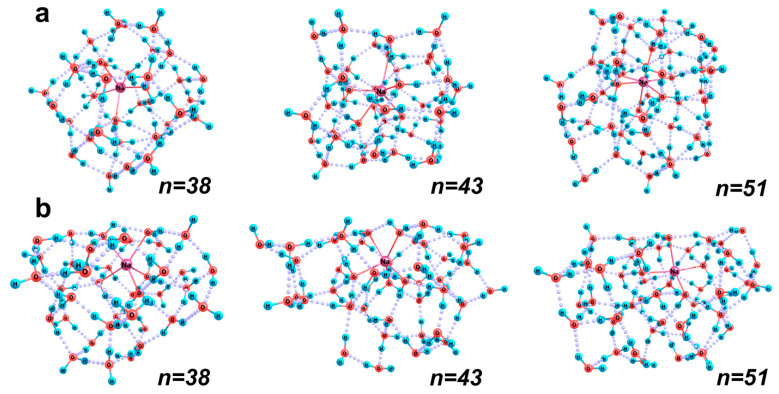
Optimized structures of the [Na(H_2_O)*_n_*]^+^ aqua complexes with *n* = 38, 43, and 51: (**a**) bulk and (**b**) surface. Na−O coordination bonds are shown with red solid lines.

**Figure 6 ijms-26-08909-f006:**
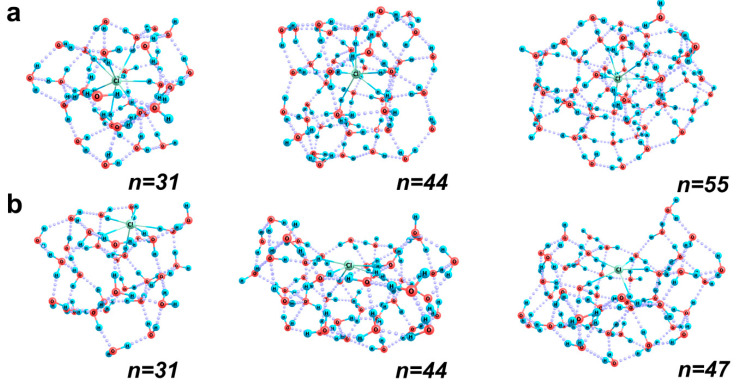
Optimized structures of the [Cl(H_2_O)*_n_*]^−^ aqua complexes with *n* = 31, 44, 47 and 55: (**a**) bulk and (**b**) surface. The Cl–O coordination bonds are shown as blue solid lines.

**Figure 7 ijms-26-08909-f007:**
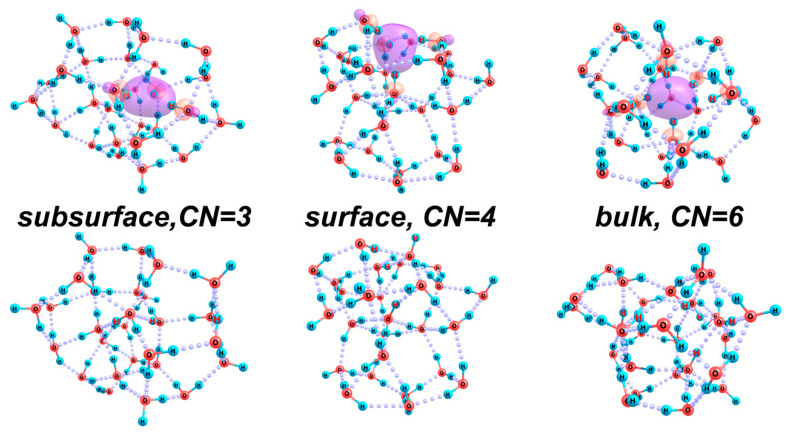
(**Top**) Optimized structures of [(H_2_O)_27_]^−^ cluster with different numbers of molecules in the first hydration shell of the excess electron (*CN* = 3, 4, 6) and different character of its localization (subsurface, surface, and bulk), the molecular orbital of which is shown with a boundary value of 0.06; and (**bottom**) relaxed structures of the water cluster upon the removal of an electron from it. The hydrogen atoms of the OH groups involved in the hydration shell of the excess electron are marked with red symbols.

**Figure 8 ijms-26-08909-f008:**
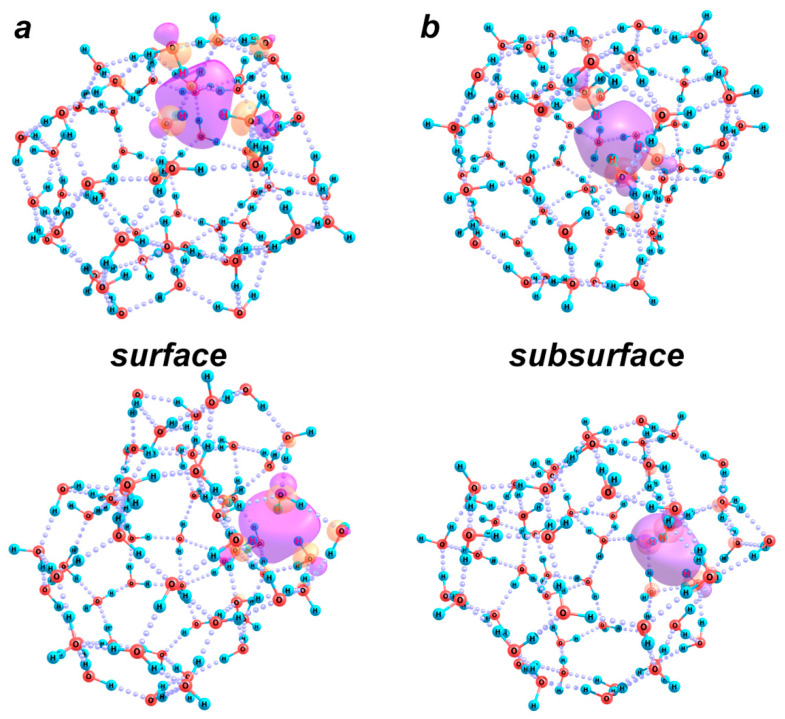
Optimized structures of (**a**) [(H_2_O)_58_]^−^ and (**b**) [(H_2_O)_60_]^−^ clusters with different characters of the excess electron localization (surface and subsurface), the molecular orbital of which is drawn with a cutoff value of 0.04. Each structure is shown in two different views. Hydrogen atoms of the OH groups involved in the hydration sphere of the excess electron are marked with red symbols.

**Figure 9 ijms-26-08909-f009:**
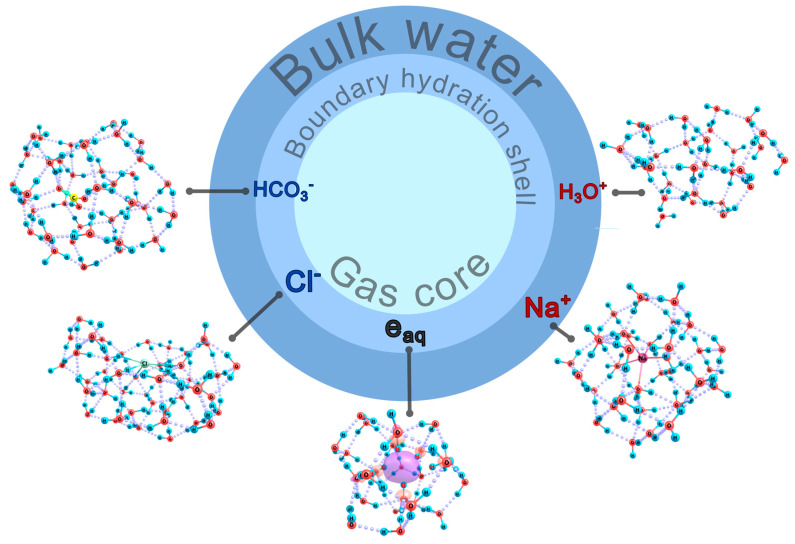
A schematic illustration of a nanobubble surrounded with boundary hydration shell in the bulk water with the domains and character of localization of different ions (Na^+^, Cl^−^, HCO_3_^−^, and H_3_O^+^) and hydrated electrons (e_aq_) shown.

**Figure 10 ijms-26-08909-f010:**
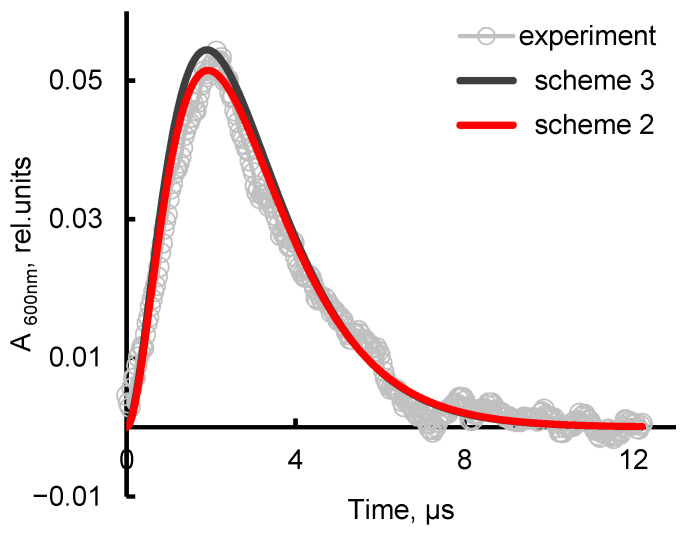
Approximation of the time dependence of the experimental spectral signal (*experiment*) recorded in 10 mg/L NaCl solution (see Figure 1) with function (3) (*scheme 2*) and function (4) (*scheme 3*). The points on the experimental curve are decimated by a factor of 8. The time of high-energy electron impact (6 MeV and 1.5-µs pulse) is set to origin.

**Figure 11 ijms-26-08909-f011:**
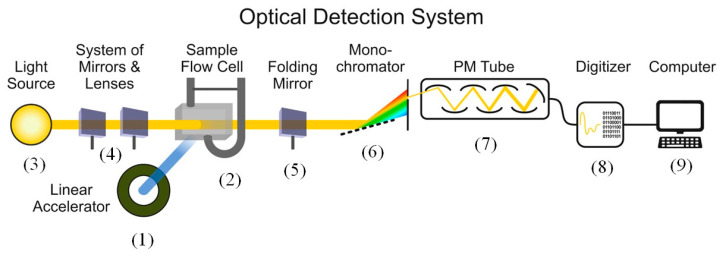
Experimental custom-made setup for measuring the absorbance of electron-irradiated liquid samples at a wavelength of 600 nm.

**Table 1 ijms-26-08909-t001:** The number of OH groups of peripheral water molecules that are not involved in hydrogen bonds in the [Na(H_2_O)*_n_*]^+^ clusters of the bulk and surface kinds.

Structure (*n*)	38	43	51
Bulk structure	13	13	14
Surface structure	11	12	12

**Table 2 ijms-26-08909-t002:** Vertical electron affinity (*VEA*) of the [Na(H_2_O)*_n_*]^+^ aqua complexes of the surface and bulk kinds.

Surface localization							
*n*	19	28	33	38	43	46	51
*VEA*, eV	1.68	1.24	1.02	1.06	1.09	0.97	1.31
Bulk localization							
*n*	31	33	38	43	48	51	55
*VEA*, eV	1.12	1.13	1.09	1.03	0.85	0.92	0.90

**Table 3 ijms-26-08909-t003:** Vertical electron detachment energy (*VDE*) of the [Cl(H_2_O)*_n_*]^−^ aqua complexes of surface and bulk kinds.

*n*	20	23	26	31	33	35	37	44	47	55
Surface localization										
*VDE*, eV	6.55	6.50	6.39	6.68	6.70	6.74	6.98	6.59	6.72	6.79
Bulk localization										
*VDE*, eV	6.88	6.95	6.74	6.68	6.83	6.78	6.77	6.84	6.91	6.95

**Table 4 ijms-26-08909-t004:** Structural and energetic characteristics of the excess electron localization in the [(H_2_O)_27_]^−^ cluster: coordination number of the electron (*CN*), the character of its localization, the relative energy (*E_rel_*) and vibrational Gibbs energy (*G_vib_*) of the cluster, the cluster reorganization energy (*E_reorg_*), and the vertical (*VDE*) and adiabatic (*ADE*) electron detachment energies.

CN	Localization Kind	*E_rel_,* kcal/mol	*G_vib_,* kcal/mol	*E_reorg_,* kcal/mol	*VDE,* eV	*ADE,* eV
3	Subsurface	3.9	4.8	39.1	22.6	−16.5
4	Surface	0.0	0.0	41.1	35.3	−5.9
6	Bulk	5.8	4.5	58.9	62.7	3.8

## Data Availability

The original contributions presented in this study are included in the manuscript. Further inquiries can be directed to the corresponding authors.

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
