# Peer review of "The Fate of Secondary Electrons in Water upon High-Energy Electron Impact: Changes in the Presence of Nanobubbles"

_ijms, 2025, doi:10.3390/ijms26188909_

Round 1

Reviewer 1 Report

Comments and Suggestions for Authors

Dear Editor,

In their paper titled “The Fate of Secondary Electrons in Water upon High-Energy Electron Impact: Changes in the Presence of Nanobubbles”, Novakovskaya et al. investigated the changes in the absorbance of secondary electrons in aqueous solutions after vibrational treatments, using both experimental methods and quantum-chemical simulations. This study follows their previous paper titled “Resonant Oscillations of Ion-Stabilized Nanobubbles in Water as a Possible Source of Electromagnetic Radiation in the Gigahertz Range.”. I recognized a strong scientific connection between the two studies. In my opinion, the paper is interesting and could be considered for publication in IJMS after major revision. The main concerns have been listed below.

  1. The term “hydrated-electron” appears in the abstract, whereas in the manuscript it is written as “hydrated electron.” Please use a uniform form for this term throughout the paper. Moreover, this term should be defined in the Introduction at its first occurrence.
  2. It would be helpful if the authors, in the Introduction, introduce both cavitation nanobubbles and air (atmospheric) nanobubbles, and clearly indicate which type is investigated in this study. It appears that the authors are working on cavitation nanobubbles.
  3. From a technical perspective, a bubble with a diameter of approximately 100 nm is not considered a nanobubble (“The lifetime of a nanobubble is proportional to its squared radius, and at Rb = 100 nm is about 0.1 ms [3].”). Bubbles with a radius smaller than 100 nm are classified as nanobubbles and have collapse times on the order of picoseconds. The authors may consult the paper below to determine the lifetime of cavitation nanobubbles. (Molecular Dynamics-Based Approach for Laser-Induced Cavitation Bubbles: Bridging Experimental and Hybrid Analytical–Computational Approaches, 2025. Thermodynamic effects on nanobubble's collapse-induced erosion using molecular dynamic simulation, 2023.)
  4. In the experimental section, what is the size of the generated nanobubbles? What are the effects of electron beams on the radius and lifetime of the bubbles? What are the criteria for the generated nanobubbles in terms of size, lifetime, and probability size distribution?
  5. In Figure 1, it appears that at 97 microseconds the adsorption reaches its maximum. Please explain why this occurs.
  6. Unfortunately, the experimental section is weak and requires modification. In this section, the authors need to clarify whether the sample contains nanobubbles and describe what happens after electron beam irradiation. Figures 1–3 are only presented without the use of physical equations or laws to explain the observations. For example, if electron impact increases the size of a nanobubble, how does this occur, and based on which criteria can this conclusion be drawn? It is recommended to discuss the underlying physics of the system rather than only reporting the diagrams.
  7. In the sentence, “We know (see above and the data published in [1] that,” the authors open a parenthesis but do not close it in this paragraph.
  8. The β factors and their exponential equation require more explanation and clarification.
  9. Please combine the results and discussion into a single section.
  10. The authors claim that they simulated nanobubbles and electrons interacting with the wall of a nanobubble. However, they did not report the radius or size of the nanobubble, nor did they specify its nature (cavitation nanobubble or air nanobubble). The authors should include a schematic of the nanobubble, indicating which part was investigated in the current research. Moreover, did they consider the vapor pressure inside the bubble?
  11. In the simulation section, how did the authors account for the effects of the electron beam?
  12. In both the experimental and simulation parts, are the electrons that enter the system absorbed, or do they simply dissolve and become hydrated in the system?
  13. The conclusion should be rewritten in two paragraphs: the first should summarize the aims, methodology, and general context of the paper, as is already partially done. The second should present the results. In other words, the results should be separated from the general explanation in the first paragraph.
  14. It is unclear whether the absorbed electrons create a new type of molecule. Please answer this question by providing valid experimental or computational evidence.
  15. Are Figures 4–7 depicting nanobubbles, or are they only clusters? If they are nanobubbles, what radius was considered for them? If they are nanoclusters, how can they and their phenomena be assumed instead of nanobubbles?
  16. In conclusion, I find this paper suitable for publication; however, the experimental section requires modification and support from physical laws to explain the phenomena occurring under electron beam irradiation. Furthermore, in the computational section, the nature and shape of the nanobubbles need to be clearly stated, and it should be explained how the electron beam is considered in the system and whether it is dissolved or absorbed. Moreover, the effects of the electron beam on the radius and lifetime of nanobubbles need to be addressed.

Author Response

We are grateful to the reviewer for careful reading of the manuscript and for his comments. The manuscript was completely rewritten in accordance with the reviewer's recommendations. Below are detailed responses to the reviewer's questions. These responses are highlighted in italics.

Question 1: The term “hydrated-electron” appears in the abstract, whereas in the manuscript it is written as “hydrated electron.” Please use a uniform form for this term throughout the paper. Moreover, this term should be defined in the Introduction at its first occurrence.

Answer 1: Yes, there was one instance in the Abstract where “hydrated electron” was written as “hydrated-electron” (in a phrase that sounds like “a range typical of hydrated-electron absorption”) solely and exclusively in order to eliminate any misunderstanding when we speak about the absorption of hydrated electrons. In all other cases (including those in Abstract) it was always written as “hydrated electron”. We have changed the introductory part of the abstract and now “hydrated electron” is indicated the same way throughout the manuscript. In addition, in the introduction we note that there are two main states of electrons in water: wet and hydrated ones. The phrase is as follows: “Such electrons, being initially quasi-free and delocalized in the conduction band, are at first transferred into an intermediate state (the so-called wet electrons) within about 110-180 fs, and, in another 240-540 fs they become completely solvated, i.e., belong to the localized state [8–13], and referred to as hydrated electrons.” (page 2).

Question 2: It would be helpful if the authors, in the Introduction, introduce both cavitation nanobubbles and air (atmospheric) nanobubbles, and clearly indicate which type is investigated in this study. It appears that the authors are working on cavitation nanobubbles.

Answer 2: As noted in our previous work [Bunkin Nikolai F., Novakovskaya Yulia V., Gerasimov Rostislav Y., Ninham Barry W., Tarasov Sergey A., Rodionova Natalia N., Stepanov German O., “Resonant Oscillations of Ion-Stabilized Nanobubbles in Water as a Possible Source of Electromagnetic Radiation in the Gigahertz Range”, International Journal of Molecular Sciences, 2025, vol. 26, № 14, 6811. https://doi.org/10.3390/ijms26146811], when water or an aqueous solution is subject to shaking, the density distribution inside the sample becomes non-monotonic. This is due to local breaks in the continuity of the liquid during shaking; the cavities formed in the domains of discontinuity are filled with gas dissolved in the liquid, which leads to the appearance of the bubble phase. The shaking process can be considered as an analogue of hydrodynamic cavitation, i.e. bubbles formed as a result of shaking can indeed be considered as cavitation nanobubbles. We have additionally quoted a monograph on hydrodynamic cavitation in the Introduction section. At the same time, bubbles resulting from hydrodynamic cavitation disappear when the external effect, which is the source of cavitation, ceases. As shown in our previous work (see Fig. 1 in this work), the nanometer-sized bubbles formed as a result of shaking exist during the entire dynamic light scattering experiment, they must be stabilized in some way (otherwise these bubbles would have to collapse within tens of microseconds). Our previous work was devoted to the study of the stabilization of such nanobubbles and their physical characteristics. In particular, it was shown that, as follows from the combination of electrostatics and hydrodynamics laws, nanobubbles, which contain atmospheric gases, can be stabilized by the charged particles (anions) included in the water shells of their gas cores. In the absence of any solutes, the ions that can play the stabilization role are bicarbonate ions, which appear due to the dissolution and hydration of carbon dioxide. We refer to these results in the present paper. Such nanobubbles stabilized by ions were named bubstons. It is this kind of nanobubbles that are considered in this study. We did not specifically emphasize in the text that the nanobubbles we studied can be considered as cavitation nanobubbles.

Question 3: From a technical perspective, a bubble with a diameter of approximately 100 nm is not considered a nanobubble (“The lifetime of a nanobubble is proportional to its squared radius, and at Rb = 100 nm is about 0.1 ms [3].”). Bubbles with a radius smaller than 100 nm are classified as nanobubbles and have collapse times on the order of picoseconds. The authors may consult the paper below to determine the lifetime of cavitation nanobubbles. (Molecular Dynamics-Based Approach for Laser-Induced Cavitation Bubbles: Bridging Experimental and Hybrid Analytical–Computational Approaches, 2025. Thermodynamic effects on nanobubble's collapse-induced erosion using molecular dynamic simulation, 2023.)

Answer 3: Indeed, bubbles with a size of 100 nm could be called submicron bubbles. We have been studying nanobubbles for quite a long time, and according to our previous experimental results, the radii of nanobubbles in water and aqueous solutions are about 50 - 200 nm. The sizes, stability, and lifetimes of nanobubbles depend not only on the nature of the environment they are formed in, but also on the way of their production. The conditions may be very different. These may be laser pulses, which locally transform the environment into plasma. These may be ultrasonic beams, which can cause strong local expansion and contraction of the environment. And these may be relatively mild effects such as shaking. Under extreme conditions, the local temperature and pressure gradients are very large, and one should take into account strongly non-equilibrium conditions. In the case of shaking, the situation drastically differs; and the sizes of air nanobubbles (of 50 to 200 nm) and their lifetimes were repeatedly proved in experiments. Furthermore, even ς-potentials of such nanobubbles were successfully measured (as stated in our previous paper). Because of the large volume of the present paper, we believe it reasonable to refer to the previous (recent) paper whenever necessary rather than repeat all the relevant results.

Question 4: In the experimental section, what is the size of the generated nanobubbles? What are the effects of electron beams on the radius and lifetime of the bubbles? What are the criteria for the generated nanobubbles in terms of size, lifetime, and probability size distribution?

Answer 4: It is known that the process of irradiating water with electrons can produce nanobubbles containing molecular hydrogen formed as a result of radiolysis, see, for example, Joseph M. Grogan, Nicholas M. Schneider, Frances M. Ross, and Haim H. Bau, Bubble and Pattern Formation in Liquid Induced by an Electron Beam, Nano Lett. 2014, 14, 359−364, doi.org/10.1021/nl404169a. Insofar as hydrogen has nearly zero electron affinity and promotes the orientation of OH bonds of the neighboring water molecules inside the bulk phase, it hampers the localization of excess electrons in the boundary shells of such nanobubbles. For the same reasons it is expected to hamper the stabilization of such nanobubbles by ions, by contrast to the nanobubbles formed previously in the samples upon shaking. We can also say that solvated electrons can provide additional stabilization of the latter nanobubbles similarly to the effect produced by anions. As shown in the paper, the character of the excess electron localization has much in common with that of chloride ions. Let us give some more estimates. According to the estimates, which can be found in the manuscript, the total number of electrons in a pulse is equal to N = 1.5×1012, while the volume number density of nanobubbles is 109 – 1010 cm-3, see Fig. 1 of our previous work [Bunkin Nikolai F., Novakovskaya Yulia V., Gerasimov Rostislav Y., Ninham Barry W., Tarasov Sergey A., Rodionova Natalia N., Stepanov German O., “Resonant Oscillations of Ion-Stabilized Nanobubbles in Water as a Possible Source of Electromagnetic Radiation in the Gigahertz Range”, International Journal of Molecular Sciences, 2025, vol. 26, № 14, 6811. https://doi.org/10.3390/ijms26146811], which shows the results of the dynamic light scattering experiment and illustrates the size distributions of nanobubbles. Taking into account that the volume of the liquid sample was approximately 100 cm3, we can conclude that the effect of primary electrons on nanobubbles inside such a volume should not be noticeable.

Question 5: In Figure 1, it appears that at 97 microseconds the adsorption reaches its maximum. Please explain why this occurs.

Answer 5: Figure 1 illustrates the actual change in the absorbance measured during an experiment, which includes the unaffected state of the specimen, the short-time high-energy electron impact, and the subsequent events. The onset of the electron impact in this experiment was at 94 μs and its duration was 1.5 μs. To stress this, the following phrase was added to the Figure caption: “The pulse onset of the electron impact coincides with the beginning of the ascending segment of the absorbance peak (shown in deep blue).

Question 6: Unfortunately, the experimental section is weak and requires modification. In this section, the authors need to clarify whether the sample contains nanobubbles and describe what happens after electron beam irradiation. Figures 1–3 are only presented without the use of physical equations or laws to explain the observations. For example, if electron impact increases the size of a nanobubble, how does this occur, and based on which criteria can this conclusion be drawn? It is recommended to discuss the underlying physics of the system rather than only reporting the diagrams.

Answer 6:

Of course, the studied samples initially contain a nanobubble phase. As shown in our previous work [Bunkin, N.F.; Shkirin, A.V.; Suyazov, N.V.; Babenko, V.A.; Sychev, A.A.; Penkov, N.V.; Belosludtsev, K.N.; Gudkov, S.V. Formation and Dynamics of Ion-Stabilized Gas Nanobubble Phase in the Bulk of Aqueous NaCl Solutions. J. Phys. Chem. B 2016, 120, 1291–1303, doi:10.1021/acs.jpcb.5b11103], nanobubbles arise in a liquid saturated with dissolved gas and containing an ionic component. If the liquid samples are kept in the open (to the atmosphere) flasks, liquid evaporates from the surface, which leads to the appearance of a temperature gradient along the height of the liquid sample. It can be shown that in flasks several centimeters high, which contain water or aqueous solutions of salts, the existence of a temperature gradient leads to the emergence of Rayleigh instability, i.e., to turbulent mixing of the liquid sample, see the quoted paper in J. Phys. Chem. B. This process is a kind of analogue of hydrodynamic cavitation, i.e. gas nanobubbles should arise in the process of such mixing. If the flask with the liquid sample is tightly sealed, the evaporation will stop, and the nanobubbles will eventually float to the surface and disappear, see the paper in J. Phys. Chem. B. Therefore, we can assume that nanobubbles are the result of hydrodynamic cavitation, but this cavitation occurs so slowly that the gas cavities formed as a result of hydrodynamic cavitation have time to be stabilized due to the adsorption of ions, i.e. such cavities turn into bubstons. Intensive vibrational treatment leads to an increase in the volume density of nanobubbles, see our previous work [Bunkin, N.F.; Novakovskaya, Y.V.; Gerasimov, R.Y.; Ninham, B.W.; Tarasov, S.A.; Rodionova, N.N.; Stepanov, G.O. Resonant Oscillations of Ion-Stabilized Nanobubbles in Water as a Possible Source of Electromagnetic Radiation in the Gigahertz Range. IJMS 2025, 26, 6811, doi:10.3390/ijms26146811], and also the paper in J. Phys. Chem. As was noted in the answer to the previous question, we assume that hydrated electrons can only provide some additional stabilization of nanobubbles with no effect on their size. According to the above estimate, even if such an influence does exist, the total number of electrons is so small that such an influence would most likely not manifest itself.

Question 7: In the sentence, “We know (see above and the data published in [1] that,” the authors open a parenthesis but do not close it in this paragraph.

Answer 7: We would like to express our gratitude to the Reviewer for the very thorough and careful reading. The closing round bracket was actually missed. It was added: “We know (see above and the data published in [1]) that the stronger the vibrational treatment…”

Question 8: The β factors and their exponential equation require more explanation and clarification.

Answer 8: We have added the following explanation: “The kinetics of the spectral response is shown in Figure 3, where the decay rates of the absorbance after reaching the peak value are compared for different samples. The peak was reached in 2–4 µs after the onset of the high-energy electron impact (6 MeV, a 1.5-µs pulse). At this step, to retrieve tentative quantitative characteristics of the dependences, the following points were taken into account. Most of the decay processes are characterized by first-order kinetics. The actual decrease in the absorbance with time (see Fig. 1) is well linearized in a semilogarithmic frame of reference. For these reasons, the absorbance decay was approximated by an exponential function A = aexp(-bt), where b  factors can be treated as apparent first-order rate constants. These b factors are shown in Figure 3.

Question 9: Please combine the results and discussion into a single section.

Answer 9: These two Sections (Results and Discussion) are typical of this Journal. At the same time, the actual results and their generalization are presented successfully in order to show, at first, the experimentally discovered trends; then, the theoretically found peculiarities that help in clarifying the possible nature of the nanobubble shells, their probable weaker or stronger stabilization, and the possible kinds of the localization of low-energy electrons. All these aspects are important for the interpretation of the results. Finally, in a separate Discussion Section, we join all the discovered features and suggest a physicochemical explanation, namely, possible kinetic schemes that show the differences in the sequences of state transformations of excess electrons. Thus, in fact, the results and discussion are not separated. They naturally follow each other, and the sectioning only helps to structure the discussion.

Question 10: The authors claim that they simulated nanobubbles and electrons interacting with the wall of a nanobubble. However, they did not report the radius or size of the nanobubble, nor did they specify its nature (cavitation nanobubble or air nanobubble). The authors should include a schematic of the nanobubble, indicating which part was investigated in the current research. Moreover, did they consider the vapor pressure inside the bubble?

Answer 10: It is obvious that we should not take into account the vapor phase inside the nanobubbles in this case. Indeed, since all experiments are carried out at room temperature and the nnanobubbles contain gas at atmospheric pressure (see our previous paper [Bunkin, N.F.; Shkirin, A.V.; Suyazov, N.V.; Babenko, V.A.; Sychev, A.A.; Penkov, N.V.; Belosludtsev, K.N.; Gudkov, S.V. Formation and Dynamics of Ion-Stabilized Gas Nanobubble Phase in the Bulk of Aqueous NaCl Solutions. J. Phys. Chem. B 2016, 120, 1291–1303, doi:10.1021/acs.jpcb.5b11103]), and the water vapor pressure at room temperature is lower by a factor of 0.02 than the atmospheric pressure, the contribution of the vapor phase can be ignored. As already noted in the answers to the previous questions, we are dealing with nanobubbles containing dissolved air, stabilized by ions and resulting from hydrodynamic cavitation. Radii of nanobubbles are given in text (around 100 nm). A schematic illustration of a nanobubble surrounded with a boundary hydration shell in the bulk water (where relative positions of diverse ions and hydrated electrons are also shown) is added, see Fig. 8 in the new version.

Question 11: In the simulation section, how did the authors account for the effects of the electron beam?

Answer 11: In Section 2.2, we consider (subsection c) the possible kinds of the localization of secondary low-energy electrons that appear in water or aqueous solutions upon their irradiation with the high-energy electron beam. It is those particles that absorb radiation at a wavelength of 600 nm, which is measured in our experiments. Here, we take into account that the energy of such electrons (as stated in the Introduction) can be up to 10 eV, which can be spent on the distortion and reorganization of the hydrogen-bond network of water and on overcoming the potential barrier created in the solution around nanobubbles due to the peculiarities in the arrangement of anions and cations (see Section 2.2, subsections a and b).

Question 12: In both the experimental and simulation parts, are the electrons that enter the system absorbed, or do they simply dissolve and become hydrated in the system?

Answer 12: The primary high-energy electrons cause very fast cascade processes, in which radicals and low-energy electrons appear. The object of our studies is the secondary low-energy electrons. To stress this, we have added the following explanation in Section 2.1: “The hydrated electrons were generated in the specimens as a result of the high-energy electron impact, which caused the formation of diverse radical particles and low-energy secondary electrons (see Section 4.2). As mentioned above, the secondary electrons on a half-picosecond scale become localized, or hydrated. And it is these electrons the absorbance of which depending on time was measured in our experiments.

Question 13: The conclusion should be rewritten in two paragraphs: the first should summarize the aims, methodology, and general context of the paper, as is already partially done. The second should present the results. In other words, the results should be separated from the general explanation in the first paragraph.

Answer 13: The Conclusions Section was fully rewritten. We believe that now it sounds much better and is much clearer.

Question 14: It is unclear whether the absorbed electrons create a new type of molecule. Please answer this question by providing valid experimental or computational evidence.

Answer 14: The low-energy secondary electrons are captured and localized by the particles of the environment. Under the experimental conditions of interest, these can be aqua complexes of sodium cations (see Section 2.2, subsection a), which thus become neutral Na(H2O)n complexes. Such complexes can easily (under the experimental conditions in question) lose the excess electrons to restore their cationic form [Na(H2O)n]+ (see Section 2.2, subsection a). An alternative path is the localization of the electrons by Bjerrum-kind defects of the H-bond network of water (see Section 2.2, subsection c), which leads to the formation of hydrated electrons, or in other words negatively charged aqua complexes [(H2O)n]-. These complexes can be found in either ground or excited electronic state; and these are the key structure fragments that predetermine the absorption of radiation studied in this work.

Question 15: Are Figures 4–7 depicting nanobubbles, or are they only clusters? If they are nanobubbles, what radius was considered for them? If they are nanoclusters, how can they and their phenomena be assumed instead of nanobubbles?

Answer 15: Figures 4-7 show nanoclusters, which help to clarify the character of the localization of sodium cations, chloride anions, and excess electrons in bulk water and in water layers around nanobubbles. As specially stressed in the text, the changes in the hydrogen-bond network of water caused by the addition of all these particles are local and restricted to two hydration shells around the particles. All the clusters shown involve two to three such shells. At the same time, the clusters differ in the position of the foreign particle. It can be localized in the center of the cluster, which corresponds to its hydration in the bulk water. And it can reside within the surface (subsurface) layer of water molecules, which corresponds to its inclusion in the surface (or boundary) layers of nanobubbles. All these kinds of clusters are considered in Section 2.2 (subsections a, b, and c) and the structural and energetic characteristics of the clusters enabled us to formulate conclusions about the possibility and favorableness of the inclusion of particles in the boundary shells of nanobubbles (for example, chloride ions should displace bicarbonate ions from the shells) and the possibility of the absorption of radiation by these particles (their aqua complexes) and the corresponding changes in their states. These conclusions form the basis for the kinetic schemes proposed in the next Section (Discussion Section).

Question 16: In conclusion, I find this paper suitable for publication; however, the experimental section requires modification and support from physical laws to explain the phenomena occurring under electron beam irradiation. Furthermore, in the computational section, the nature and shape of the nanobubbles need to be clearly stated, and it should be explained how the electron beam is considered in the system and whether it is dissolved or absorbed. Moreover, the effects of the electron beam on the radius and lifetime of nanobubbles need to be addressed.

Answer 16: This was done, see above.

Reviewer 2 Report

Comments and Suggestions for Authors

This is an interesting article and is acceptable after minor revision. Authors have given a detailed explanation of their research. It is advised that authors should also provide some more references in the introduction section regarding the application of this concept specially the nanobubbles for example in case of water treatment etc.

Author Response

We are grateful to the reviewer for careful reading of the manuscript and for his comments. The manuscript was substantially rewritten in accordance with the reviewer's recommendations. Below are detailed responses to the reviewer's questions. These responses are highlighted in italics.

Question 1: It is advised that authors should also provide some more references in the introduction section regarding the application of this concept specially the nanobubbles for example in case of water treatment etc.

Answer 1:

The Introduction was extended. Particularly, it is written that “nanobubbles can affect the refractive index [8] and light scattering [9] in the medium, and, if stabilized, are subject to the flotation effect, adsorbing dissolved substances on their surface [10]”.